# Evolutive 3D Modeling: A Proposal for a New Generative Design Methodology

**Jaime Nebot** , **Juan A. Peña** * and **Carmelo López Gómez**

Department of Design and Manufacturing Engineering, Engineering Graphics, University of Zaragoza, 50009 Zaragoza, Spain; 687292@unizar.es (J.N.); melopez@unizar.es (C.L.G.)
* Correspondence: juanp@unizar.es

**Abstract:** At present, traditional 3D modeling programs consist of a set of tools that reflect conventional means of mechanical manufacturing and have limitations in relation with the current manufacturing capacities. On the other hand, organic and morphing 3D modeling programs are designed to transform a model from one known shape to another also known shape. Generative design helps the designers to detach themselves during the design process and can provide them with completely unexpected geometrical solutions. In this paper, starting from 3D morphing techniques and genetic algorithms, a new methodology of product shape definition is developed, capable of imitating processes that occur in nature and aimed at creating new and different product designs. This methodology enables to overcome the limitations imposed by design fixation and allows better exploitation of the great possibilities granted by the new manufacturing techniques, most notably additive manufacturing. The initial process of research and information gathering gives this work a solid basis to develop the new methodology. The results of this initial process are briefly resumed in this paper in order to explain the main motivation for developing this work. The workflow of this methodology is presented as a theoretical process, since its implementation has not been, at least for the moment, put into practice. Before presenting the conclusion for this proposal, several examples have been formulated in order to help the reader to catch the point of the entire process.

**Keywords:** evolutive modeling; generative design; design fixation; 3D morphing; genetic algorithms; geometric modeling; 3D modeling

## 1. Introduction

Design fixation deprives designers of a great number of product solutions, thus affecting the development of the world that surrounds us. For this reason, it is necessary to break with the preconceived figures and geometries found during the current design processes, and to find new ways to create structure and shape of products. By doing so, we manage to free the design process from any historic, cultural or conventional technique influence.

With the emerging manufacturing techniques, especially with additive manufacturing, the versatility of product shape is considerably increased, and we should take full advantage of all the potential that these new possibilities offer, breaking with the fixation that exists in the design process when it comes to generating new shapes [1].

In addition, most current 3D modeling software includes design tools geared towards conventional manufacturing. This also implies a diminishing of design freedom when modeling new products.

In this paper, a new way of facing up to the object modeling and product definition needs is developed. A new 3D modeling methodology is proposed, based on generative design, in order to help overcome creative fixation derived from the limitations of conventional manufacturing techniques. Using 3D morphing tools and genetic algorithms as a basis, a system is developed in which, by imitating the processes of natural evolution, we

obtain a series of possible solutions that are different from those that would be reached by designers just through their own creative means.

In previous studies, such as the one developed by the authors of [2], a generative system is used in which several results are obtained starting from the initial definition of the general features of a product. In order to obtain these results, different geometric primitives, located in different points of a 3D space, are combined by using operators such as patterns and symmetries.

On the other hand, the authors of [3] proposed a generative method that uses traditional CAD technology and genetic algorithms. A base model is created and, by applying random mutations to it, several new models are generated, which will be finally accepted or not according to a series of requirements.

In both cases, the techniques used for the shaping of 3D models do limit the results. The reason is that geometric primitive modeling and traditional CAD techniques do not grant as much shaping freedom as polygonal mesh modeling, which enables more organic results.

Accordingly, the great advantage of the present proposal over previous studies is the use of a much more abstract initial model and far more organic evolution. This means that the variability of the solutions that are compatible with the product specifications is much larger.

Even so, the most notable advance that distinguishes the evolutionary techniques developed so far from the methodology presented in this paper is that the shaping of our products is purely determined by their functional requirements. Furthermore, the evolutionary process we propose is cyclical and iterative, producing far more developed solutions. Consequently, designers can detach themselves during the design process, which is the main goal we seek.

*Working Method*

An early stage of search, gathering and analysis of information has provided the necessary basis to establish a solid methodology.

Since this paper sums up the process of creating a new methodology from the beginning, an important part of the information gathered has been used to define our fields of action, while another important part aimed at the enhancement and reinforcement of our proposal. Other parts focused on how creation and evolution occur in nature, while staying completely free from any kind of influence or fixation.

Once all the information was analyzed and filtered, it was used to help to build the new methodology. By taking some of its contents and structuring them adequately, every step of the proposed process was composed. Several alternatives were studied, compared and assessed, until the final definition of this methodology proposal was attained.

The results of this process intend to reach the highest possible abstraction level in order to increase the detachment from how a designer would solve the same problem. Therefore, all the considered options to be implemented were assessed according to their level of abstraction, although their feasibility in terms of current and near future computation capabilities was always taken into consideration.

Three examples have been proposed in order to better explain the process, since it is, for the moment, theoretical. Two of them have been chosen according to their similarities to the process that is being presented and the third one tries to help understand what exactly we are looking for in this research.

## 2. Current Works

### 2.1. Design Fixation

The term "fixation" in itself, defines an obsessive interest in a person or object. Based on this term, different concepts applied to more specific situations have evolved.

In [4], the term "functional fixedness" is coined. This term refers to the fixation on just one function of a product and, as a consequence, to the mental blocking of any

creative reinterpretation of the functions of objects that are familiar to us. Similarly, ref [5] introduces the term "mental-set" or Einstellung effect, referring to a situation in which an individual focuses his/her attention on just one particular process, acting as a block to find different ways to solve a problem.

Finally, the term "design fixation" refers to the fixation that can be found in the field of design. In [6], a design exercise is proposed to a group of designers in which, as part of the briefing, examples of solutions are included. The result of the inclusion of the examples is that the capacities of abstraction of the designers are reduced during the design process. This is due to the fact that they unconsciously incorporate into their products properties of the given examples. In this article, the term "design fixation" is used to describe a blind adherence to a limited set of ideas during a design process.

As we can see, design fixation has been a case of study for many years and, although it has been applied to different fields and different definitions have been adopted for it, we always encounter, as a common aspect, the fact that it is considered a negative feature. For this reason, the aim of most existing studies is to limit or to correct this phenomenon [7]. Despite having been a case of study for almost a century, there is still much to research regarding this subject [8].

### 2.2. Morphing

The art of representing shape transformation or morphing has been studied for many years. Currently, 3D morphing techniques are widely used in computer graphics, and we can find them everywhere. In fact, its area of application is large, ranging from special effects animation in the film industry to simulation and representation of medical and scientific processes [9].

The breakthroughs in the morphing techniques that have been made so far are mainly focused on the optimization of the process itself, improving the visualization of the transformations and developing interpolation and problem-solving techniques. Basically, what is being studied at present are the paths that the different geometric elements have to follow in order to change from an initial state to a final state of a figure, aiming at a pleasant visualization of the transformation [10–12].

Thus, we are dealing with a closed process, in which the initial and final shapes are previously defined and stay unchanged during the process, the main action being the transformation from one to the other.

### 2.3. New Manufacturing Techniques

Presently, a great number of new manufacturing techniques are emerging. These techniques overcome some of the geometric limitations of industrial products, formerly imposed by the traditional manufacturing techniques. Among these new techniques, additive manufacturing stands out.

Additive manufacturing can be defined as the technique of adding material, layer by layer, to obtain the desired shape. It is in complete contrast to traditional mechanization, in which, basically, material is removed from an initial volume. Another traditional manufacturing technique is the use of molds for filling with melted material.

Unlike the traditional mechanization (lathe and milling cutter) where the plane and the cylinder are the basic shapes worked with, additive manufacturing allows one to create almost any geometry due to its working principles.

The paradigm of additive manufacturing enables a more complex object shaping in a much easier way [13]. In addition, it gives us the possibility to create shapes that are completely unattainable with conventional manufacturing methods, which makes this new technique revolutionary.

As explained in [14], additive manufacturing provides an increase of design freedom, apart from allowing a direct transformation of 3D model files into completely functional products.

Accordingly, we can conclude that the capacity to physically reproduce a 3D model has considerably increased. Furthermore, these new manufacturing techniques adapt well

to the new market trends, with productive aspects such as short series, custom series, exclusive series, etc.

### 2.4. Genetic Algorithms

To notice and to get inspiration from different elements found in nature in order to produce artificial products is a common practice that has been constantly being carried out throughout human history. This is what we call bioinspired design. One of the several branches of bioinspired design is bioinspired computation, to which genetic algorithms belong [15].

A genetic algorithm is one that applies the bases of natural evolution to computation by imitating it [16]. It has previously been used to develop methodologies of generative design, which consists of the automatic generation of different formal solutions using computational calculus. These methodologies exclusively create 3D shapes without taking the intrinsic criteria of the product into account. For this reason, they do not generate viable solutions on their own.

Genetic algorithms work in such a way that an initial population of samples interacts with itself, thus creating a new generation of samples, of which only those that comply with a series of requirements survive. The more times the process is repeated and the more samples are used, the more precise the results tend to be.

Genetic algorithms can be used for concept design exploration [2]. This is due to the fact that, in the process of conceptualization, they are capable of providing solutions that a designer would not be able to conceive due to his/her design fixation, among other factors.

In contrast, working with genetic algorithms requires a high computational workload, which makes it necessary to correctly select the initial parameters so that the volume of necessary resources needed for the development of the simulation does not shoot up.

## 3. Proposal

### 3.1. Background

Currently, 3D printing gives our products a lot of geometric freedom in comparison with conventional means of manufacturing, but it has been found that, due to fixation, these new advantages are not being exploited as they could be.

On the other hand, we find that interesting studies have been carried out, separately, in the fields of genetic algorithms and 3D morphing, but never relating them to each other.

For this reason, we decided to initiate a new branch of research by joining these series of concepts and techniques and developing them together as a new design and 3D modeling methodology. The aim of this new approach is to detach the designer from the preconceived shapes inherited from culture and conventional means of manufacturing.

### 3.2. Conceptual Explanation

Currently, conventional 3D morphing processes consist of changing from an initial to a final shape while having a clear mental image of how the latter should be. This means that we are dealing with a linear process between two models we already know. In contrast, the methodology defined in this paper allows, starting from a known initial model, to generate a variable number of suitable solutions. In other words, we are proposing a process of generative conceptual creation which is divergent, and in which the designer, once in possession of a series of results, will just have to top off.

This work presents a new paradigm in 3D modeling techniques, consisting of the creation of different geometric solutions to a problem starting from a series of requirements and stimuli.

In order to implement this process, the working structure of genetic algorithms, as proposed in [16], is used. In this way, the defined process is a perfect analogy to the genetic evolution of the species of living beings.

Since the methodology proposed combines the evolutionary processes and 3D object modeling, it seems appropriate to term it "evolutive modeling".

Evolutive modeling consists in the creation, by means of genetic manipulation of geometries in a 3D space, of models capable of representing an object in real life. The conditioning factors that will define the final shape will be, on the one hand, design requirements and, on the other hand, environmental factors.

We start from the volumetric definition of one or several model objects, which will evolve throughout a variable number of generations. During this process, as the different generations follow each other, different changes in the structure of the model objects take place, induced by the environmental factors previously mentioned. Likewise, object models will be filtered each time according to the also previously mentioned design requirements.

In conclusion, the results obtained in this process will be regulated, first, by the design requirements, attending to the functional needs of the product, and second, by the environmental factors, attending to its expected working environment.

### 3.3. Proposed Methodology

The evolutive modeling process is divided into three stages. In the initial stage, all of the factors that are necessary to obtain suitable results will be determined. The second stage consists of the development of the evolutive process, where different generations of solutions will follow each other according to the posed requirements and factors. In the last stage, once the results are obtained, the designer will turn these into one or more final models by adding any necessary finishing touches.

Strictly speaking, the purely evolutionary aspects will only happen in the first and second stages, but it is important to point out that a postproduction (third stage) is necessary because of the possibility that more than one result is obtained and that these results may not be completely finished.

In order to explain the different elements that contribute to the correct performance of the process, we use the correspondence between concepts from nature and concepts of evolutionary computation, as proposed in [16]. See Table 1.

**Table 1.** Correspondence of evolutionary concepts in nature and computation.

| Nature | Evolutionary Computation |
| :---: | :---: |
| Individual | Solution to a problem |
| Population | Collection of solutions |
| Fitness | Quality of solutions |
| Chromosome | Representation of a solution |
| Gene | Part of a representation of a solution |
| Crossover | Binary search operator |
| Mutation | Unary search operator |
| Reproduction | Reuse of solutions |
| Selection | Keeping good solutions |

Source: [16].

### 3.3.1. Stage 1: Initial Parameters

The elements that will determine the results at the end of the process are the initial parameters.

In the first place, it will be necessary to establish an initial population of individuals. The first decision to be taken will be if just one or several initial individuals are needed. In case of using just one individual, the evolutionary process will be more resource-consuming, and it will have to be divergent in the first generations. In case of using an initial population of several individuals, the process will be less resource-consuming, but, depending on the degree of their geometric definition, we risk a loss in the level of abstraction of the results. Obviously, another important aspect to bear in mind is the topology of the initial model or models, as they must be suited to undergo different morphological modifications.

We have to determine the design requirements in order to steer the process towards suitable results. At this point, we can include both design specifications taken from the essence of the product, as well as limitations to help the process keep in track towards

valid results. The more requirements and limitations applied, the less degree of abstraction obtained.

It is important to establish what the different properties that we want to determine our products are, and, therefore, it is necessary to define their purely intrinsic aspects.

Next, the environmental factors are taken into account. That means including different conditioning elements into the process that represent the situation or scene in which our products will develop their activities or functions during their service life. We help the results to be more suitable for developing their activities in a specific environment or situation.

This way, what we do is to determine the function that the product will carry out and under which conditions. From this point on, models will evolve towards solutions that will fit the imposed requirements and adapt better to the environmental conditions.

Finally, the termination criteria of the process will have to be determined, be it by limiting the number of generations, be it by determining a specific situation in which the process should stop. In the second case, each generation will have to be analyzed and assessed according to the defined parameters.

The key point at this stage will be to find the perfect balance between abstraction and computational workload. Starting from absolute abstraction, which comes to imitating pure genetic evolution, the more requirements and conditioning factors we include, the more guided the process will be, and, as a result, the computational workload needed will be less. In return, the level of abstraction of the process, and, consequently, of the results, will obviously be reduced.

### 3.3.2. Stage 2: Evolution

This stage focuses on the evolutive development in itself, that is to say, on the genetic algorithm. Depending on different aspects of the evolutive process, we will be able to obtain more or fewer results, and more or less precise.

Once the initial population of individuals, the requirements and the conditioning factors are determined, the following process will be applied to each generation (Figure 1):

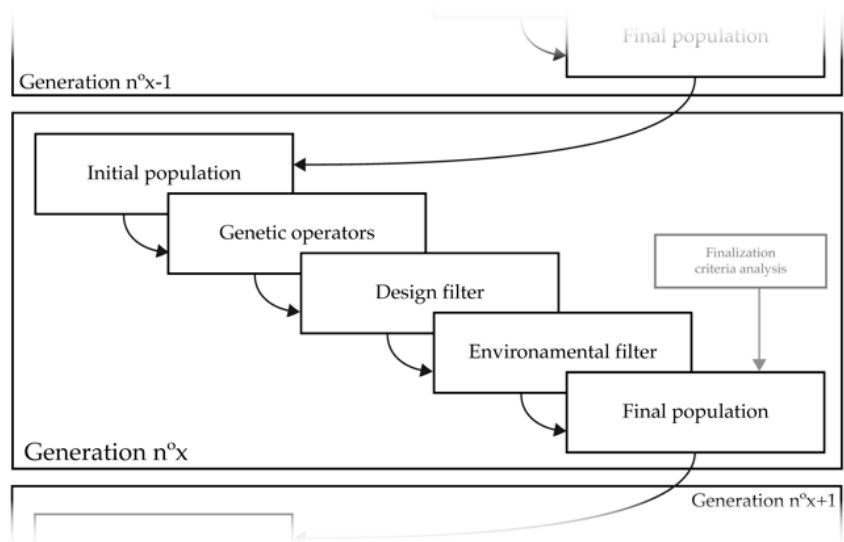

**Figure 1.** Creation of a new generation process from the previous one.

Firstly, with the aim of obtaining a more diverse population than the previous one, three genetic operators (reproduction, crossover and mutation) will be applied to each individual.

These three genetic operators are the ones proposed in [16]. The reproduction operator consists in maintaining individuals throughout the generations, which is important in order not to lose the good solutions that may have appeared in the first generations,

and also to make it easier for the individuals to cluster in the last part of the generative process. Crossover, on the other hand, consists of joining two different individuals in order to generate new ones, which include properties inherited from both predecessors, thus exploring new feasible solutions within a defined frame. Finally, mutation is a unary operator that randomly alters different properties of an individual. This operator initiates new evolutive paths, being the only one capable of diverging from one single element. This is especially important when the initial population of a process is made up of just one individual.

The next step is to filter the current population according to the design requirements— that is, we are controlling the population volume by following the limits established by the designer. So, we do not spend computational resources on solutions that may be discarded later on.

Then, a second filter is applied, in which different environmental conditions are simulated with each individual, assessing its degree of suitability according to the required functionalities. On the grounds of this assessment, those individuals that perform best will continue in the process, while the others will be rejected.

It is important to point out that, in both of the filtering steps explained, the selection process does not need to be binary. This means that individuals, even if they do not completely fulfill the requirements, can be accepted, thus allowing the exploration of other paths with the potential to become a perfectly valid solution.

Finally, once the two filtering steps are finished, we obtain the population, which constitutes the initial population of the following generation.

This sequence can be repeated as many times as needed, but it is important to find the balance between the number of individuals and the diversity of solutions. Therefore, we will have to play with several factors: grades of permissiveness during filtering, maximum and/or minimum number of individuals, grades of randomness, variability of the genetic operators, etc.

The evolutive process aims to be nonlinear along generations. The way forward, in this second stage, is to go through three different blocks of evolution. First is a divergent block, in which the priority is a large and scattered generation of individuals, in order to reach a greater diversity. In the next block, a branching trend is intended, where several groups of similar individuals cluster, forming different branches, each of them with common properties. The last block is intended to stabilize the different branches, each of them constituting a result at the end of the process. This means that the parameters given for the development and filtering of each generation may depend on the progress of the specific evolutive process, and can, therefore, be variable.

### 3.3.3. Stage 3: Results

The main goal is to obtain a final population with a great diversity of solutions, but in which the individuals are clearly grouped in an evident way. In doing so, we will obtain different results, yet strongly consolidated, due to the fact that the more individuals of the same branch survive, the more solid the solution will be.

In conclusion, at the end of the process, what we get is a series of groups of solutions, each of them constituting a final result in itself.

From this point onward, it is up to the designer to make the most of the obtained results to specify a definite concept and to develop the final product.

## 4. Results

### *4.1. Practical Examples*

Each of the examples shown in this chapter serves to illustrate different parts of the proposed process.

In the first example, using a tree frog as a model, we will be able to appreciate how individuals evolve and how the genetic operators work and are represented in 3D models.

In the second example, we will show how the filtering process of individuals in each generation will be carried out. As a model, we will use male deer horns in two different stages of growth.

These specific nonartificial examples have been chosen due to the fact that they represent natural beings that are completely free from any kind of fixation. Since they are a product of mutation and natural selection, they are the perfect example to explain the topic we are dealing with in this paper.

Anyway, a further example has been presented in order to help figuring out a specific implementation of this methodology. In this case, the inputs and the expected outputs are the main aspects we put the focus on.

### 4.1.1. Example of the Generation of New Individuals

To illustrate the different operations that will take place throughout the process, we are going to use the analogy of the metamorphosis of a tree frog. We use this example because, in nature, we can observe all the phases this animal goes through, from the embryonic state to adulthood.

We start, as a base model, from a state in which the frog is just a trunk (Figure 2). From this point on, new individuals will begin to be generated (Figure 3), as a product of the different genetic operations (Figure 4). Each of these individuals will be assessed according to the sought objectives. In this way, the individuals that do not fulfill (or fulfill less) the requirements will be eliminated from the process.

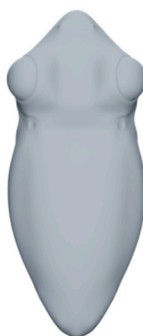

**Figure 2.** Initial individual of the simulation.

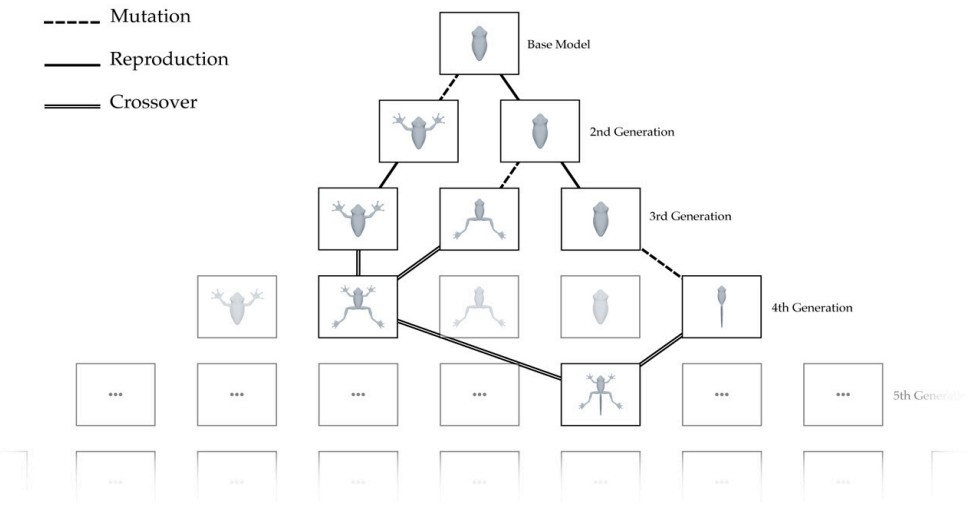

**Figure 3.** Example of the evolution of individuals in the first generations of the process.

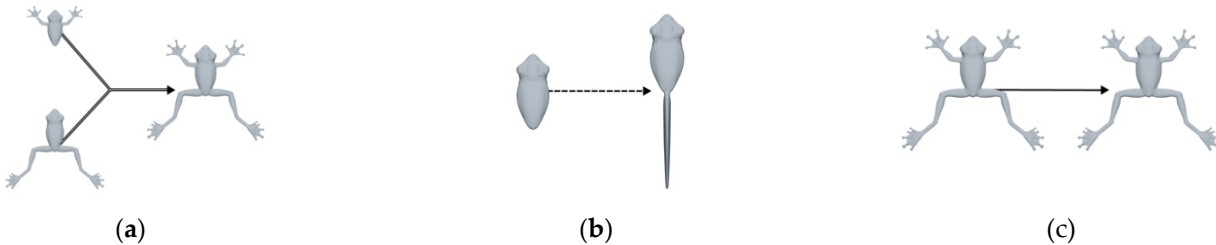

**Figure 4.** Genetic operators: crossover (**a**), mutation (**b**) and reproduction (**c**).

The reproduction operator can generate up to one new individual from each individual of the previous generation. The crossover operator generates

$$n(n-1)/2 \tag{1}$$

new individuals in the next generations, but only in the case that both parents contribute with the same percentage of genetic heritage. Otherwise, the capacity of creating new individuals could increase towards infinite. The results obtained from the mutation operator could also be, theoretically, infinite.

The aim of the evolutive process illustrated in this example could be to find the model that best moves in a liquid environment, such as water. Therefore, each of the individuals of each population will undergo an analysis in a simulation in order to assess to what extent it fulfills the aim, thus finding and promoting the best possible solutions. Obviously, this specific simulation would require a great quantity of parameters and computational workload.

The models shown in Figure 5 are an example of the results that could have been obtained in this specific case.

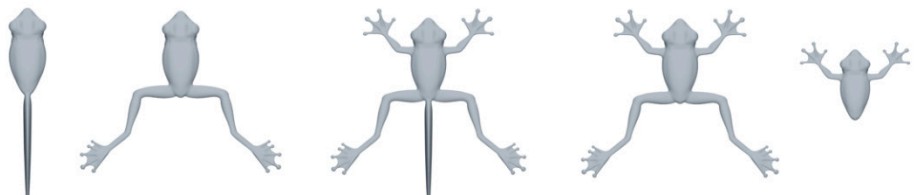

**Figure 5.** Possible results obtained at the end of the process, when looking for a frog that best moves in a liquid environment. The individual on the left the best moving and the individual on the right the worst moving.

### 4.1.2. Example of Generational Filtering

In the case of wanting to design a model which receives the smallest damage possible when colliding with an equal one, let us imagine the simulation we should run in order to filter each generation. As an example, we will use the horns of a male deer in different phases of growth.

Apart from defending themselves from predators, male deer horns are designed to hurt each other the least while fighting an equal. This is a way to keep a high number of healthy males alive, despite their recurrent fights for the control of the territories or the herd.

Applied to our genetic process, the need of two deer not to hurt themselves while charging each other, is reflected as a design requirement. This requirement will be used for filtering a generation.

On the other hand, a good example of an environmental factor, in this case, could be the climate of the natural environment in which the horns will be colliding, since, depending on whether it is wetter or drier, the friction factor between the two pairs of horns can vary.

When it comes to filtering, the first step would be to carry out the simulation. Then, the results should be assessed according to the requirements in order to detect which individuals perform best.

In this example, the simulation would be the frontal collision (with certain degrees of variability) of two pairs of identical horns (Figure 6). This simulation should be performed several times with movements starting from different initial spots. In the case they should get entangled and not reach the limit where the head of the opposing animal is supposed to be, the result would be satisfactory. Six simulations were carried out using the same horn models in order to analyze their performance (Figure 7).

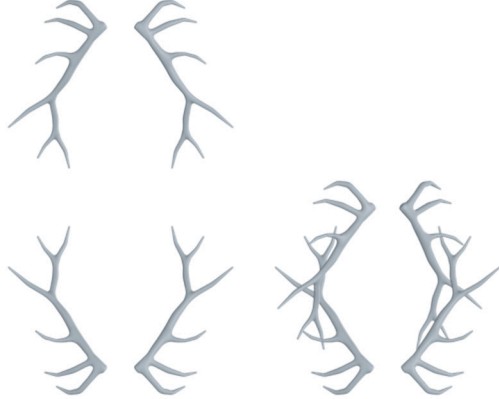

**Figure 6.** Horn impact simulation.

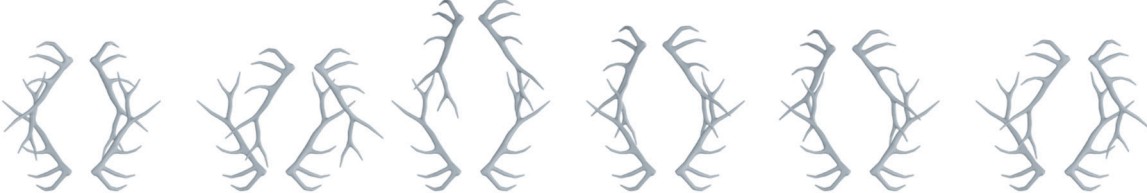

**Figure 7.** Several horn impact simulations.

From carrying out the same process with less developed horns (Figure 8), we can see that two of the simulations have not been satisfactory, as the established limit has been exceeded. This means that the more developed pair of horns would be a better option according to the assessed requirement, so that it would have more chances to keep evolving and to promote into the following generations.

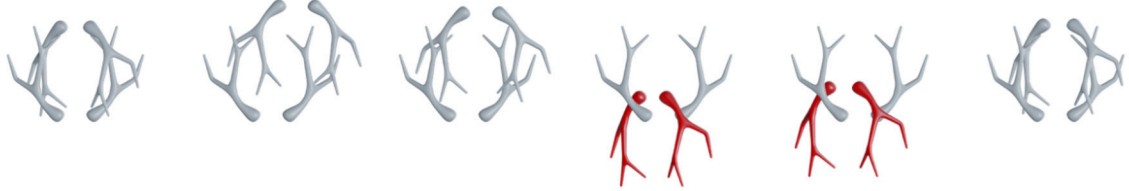

**Figure 8.** Several less developed horn impact simulations. Two of them (4 and 5) result in a negative performance according to the requirements.

4.1.3. Example of Specific Implementation

In order to explain how a realistic use of this method would be, the generation of a solar collector is proposed. The objective of the collector could be to collect the most quantity of light using the less surface possible.

The first thing to do would be to establish the initial population, introducing several 3D models, trying to keep the highest level of abstraction (for example, using primitive

shapes, or only a sphere). Then, the design requirements should be defined. Limitations such as maximum height or weight would be suitable. Next, we would have to define the parameters that will rule the environmental factors simulation. In this case, we should take every single element that may affect the product in its final working location, such as weather, latitude, humidity or vegetation, into account. After all these inputs, we will have to define our evaluation criteria, which, in this particular case, would be to get the most amount of light using the less surface possible.

A maximum number of individuals, iterations or genetic operations could be used in order to modulate the duration of the entire generative process. Directly related to this duration, the accuracy and, as a consequence, the level of development of the results, will also be affected.

Once the initial parameters are set, the process would run independently, according to the given specifications. The result would be a collection of solutions that should inspire the designer to define de final product.

## 5. Conclusions

### 5.1. Discussion

More than a proposal for a new approach to or a new development within the limits of the techniques mentioned in the "state of the art" section, the contribution of this work is the establishment of a new methodology of generative design, in which, by applying techniques such as morphing and genetic algorithms to an initial 3D model, we obtain a series of concepts that will help us to develop new products capable of being manufactured, overcoming to a great extend the problem of design fixation.

As a starting point, we take these two techniques (3D morphing and genetic algorithms) due to the fact that they are the computational processes the most similar to natural genetic evolution. The intention is to imitate the randomness and the singularities that, over the time, have been shaping a world in which everything is in perfect balance. Thus, the aim is to find products that will adapt to and take advantage of their environment, trying not to alter it and looking for a completely functional balance.

In this manner, there is the possibility to create new products that do not alter their ecosystem but adapt to it, being easier to manufacture, and being completely functional.

Although the main objective is to obtain purely functional models, it is clear that, at least for the time being, the final intervention of a designer will be necessary in order to complete the product definition—that is to say, at present this methodology would mean a great help for the process of conceptual abstraction, but we cannot guarantee that a completely finished model will be generated in all cases. In addition, as, for the moment being, we do not take into consideration the aesthetic aspects of the products in our methodology, we understand these as a separate task to be faced after the results are obtained.

### 5.2. Definition of "Evolutive Modeling"

Evolutive modeling is a product design methodology based on the structural modeling of 3D shapes by means of genetic techniques, considering the functionality of the product and the environment in which its activity will take place.

### 5.3. Proposals for the Future

The concept of evolutive modeling must be further developed with the aim of turning into a new paradigm or trend in product design and 3D modeling, more in accordance with the current and future manufacturing capabilities.

The 3D evolutive modeling methodology proposed here will have to be implemented in a series of computer applications in order to automate the process and make the creation of new products much easier. These applications will be an important part of the projects still to come and will be used as tools for further experimenting and testing.

Generative design minimizes the restrictions in the product design process imposed by design fixation that arises, to a great extent, from conventional manufacturing techniques. The final objective of the new line of research initiated by this proposal is to develop new 3D modeling techniques in the field of generative design that are efficient and effective, bearing in mind the capabilities of the new and future manufacturing techniques.

**Author Contributions:** Conceptualization, J.N., J.A.P. and C.L.G.; Formal analysis, J.N., J.A.P. and C.L.G.; Funding acquisition, J.A.P. and C.L.G.; Investigation, J.N., J.A.P. and C.L.G.; Methodology, J.N., J.A.P. and C.L.G.; Project administration, J.A.P.; Resources, J.N., J.A.P. and C.L.G.; Supervision, J.A.P. and C.L.G.; Validation, J.A.P. and C.L.G.; Visualization, J.N.; Writing—original draft, J.N.; Writing—review & editing, J.A.P. and C.L.G. All authors have read and agreed to the published version of the manuscript.

**Funding:** This research received no external funding.

**Institutional Review Board Statement:** Not applicable.

**Informed Consent Statement:** Not applicable.

**Data Availability Statement:** Not applicable.

**Conflicts of Interest:** The authors declare no conflict of interest.

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
