# Peer review of "Evolutive 3D Modeling: A Proposal for a New Generative Design Methodology"

_symmetry, doi:10.3390/sym13020338_

Round 1

Reviewer 1 Report

Overall

This paper presents the development of a new methodology for evolutive 3D modelling, based on morphing and genetic algorithms.

Abstract is clear and well structured, presenting almost all the elements needed (introduction to topic, main areas addressed, main results). Lacks methodology.

Keywords are adequate.

It is the reviewer opinion that the paper is well written, in a clear manner, helpful to understand the subject. Is original and presents a step forward in the area.

The paper is structured in a simple way, presenting a direct relation between the work developed and the paper chapters. But could be improved by giving autonomy to chapter 1.1.

Lacks a chapter presenting the methodology of the research developed underlaying this paper. It’s only presented the working methodology for the evolutive modelling.

  1.  

This chapter presents a clear introduction to the topics and the work developed.

Sub-chapter 1.1. should be transformed into chapter 2, renumbering the following chapters.

  1.  

Ok.

3.

Could present an example of an artificial (man designed) object like a chair, so that the reader, namely a designer, can relate and understand the potentiality of the methodology.

  1.  

This last chapter is presented in the form of discussion, but should be the conclusion (since it also integrates recommendation for next steps)

References

The paper presents a robust set of up to date references.

Author Response

Thank you for your review. The response is attached as a pdf.

Reviewer 2 Report

The manuscript is devoted to the development of a technique of the product shape definition using the evolutive 3D modelling method.  The research topic is very interested and actual.

  1. The abstract should be improved. It should briefly reflect the contains of the research and obtained results.
  2. The section "Current works" is absent. Of course, the authors have noticed some works in this subject area in the Introduction section, however, to my mind, it is not enough.
  3. It will be better if, after the current state of works analysis, the unsolved part of the general problem will be allocated. Then, the objective of the research will be formulated considering the unsolved part of the problem.
  4. The topic of the paper contains the phrase "A New Generative Design Technique Proposal ". I think that it is not correct in terms of English grammar. Moreover, I have not seen this new generative design in the paper. The section "Materials and Method" contains much information but I have not seen the concrete algorithm or block chart of the authors' techniques.
  5. The “Discussion” section also does not reflect the discussion of the obtained results with an analysis of both the advantages and shortcomings. I think, that this section should be rewritten by adding the Conclusions section.
  6. The list of the references should be extended too by adding the current works in this subject area in the section "Current works".
  7. To my mind, the style and English grammar of the manuscript should be modified too.

Author Response

(The authors gave the same response as above.)

Round 2

Reviewer 2 Report

Tanks for corrections,

I have not more questions